# Supply chain concentration and enterprise financialization: Evidence from listed companies in China's manufacturing industry

**Huanhuan He**[☯], **Zongwen Zuo**[iD]*[☯]

School of International Trade & Economics, Anhui University of Finance & Economics, Bengbu, Anhui, PR China

☯ These authors contributed equally to this work.
* 120081077@aufe.edu.cn

**Data Availability Statement:** All relevant data are within the paper and its Supporting information files.

## Abstract

Enterprise financialization will block the equipment update and technological innovation of enterprises by crowding out the main business funds. The risks and benefits of supply chain concentration will affect the enterprise financialization. This paper selects the panel data of A-share listed companies in China from 2009 to 2021, and uses fixed effect regression to analyze the impact of supply chain concentration on enterprise financialization. The conclusions show: both suppliers and customers concentration significantly promote the financialization of enterprises, and this conclusion is still valid after a series of tests; This kind of financialization effect is heterogeneous in four aspects: the nature of property rights, the scale of enterprises, the intensity of industrial competition and the level of economic development in the region where the enterprises are located; the mechanism analysis show that customer concentration can affect enterprise financialization through upstream commercial credit, but supplier concentration cannot affect enterprise financialization through downstream commercial credit.

## 1 Introduction

The 20th National Congress of the Communist Party of China clearly pointed out that we should focus on the real economy, promote new industrialization, and accelerate the building of a manufacturing power. The manufacturing industry plays a leading and supporting role in China's modern economic system. As the pillar of the real economy, the manufacturing industry is characterized by a long value chain, strong relevance, large driving force and wide radiation range. Therefore, the in-depth development of the manufacturing industry is of great significance for improving the industrial structure, optimizing the allocation of resources and speeding up the transformation of old and new drivers. However, in recent years, due to the overcapacity of the real economy, the rate of return on investment of manufacturing enterprises has declined significantly. So, enterprises allocate a large amount of capital into the financial system [1]. Specifically, the average value of financial assets held by enterprises in the CSMAR database was about 1.3 billion yuan in 2009, 2.1 billion yuan in 2016, and nearly 2.8 billion yuan in 2021. It can be found that the financial scale of Chinese enterprises is

**Funding:** Natural Science Key Program Fund for Anhui Provincial Colleges and Universities (No. 2022AH050593); Anhui University Student Innovation and Entrepreneurship Project "Research on the Impact of Supply Chain Concentration on Enterprise Finance: Based on Empirical Evidence of Listed Companies in China".

**Competing interests:** The authors have declared that no competing interests exist.

expanding. Although financial investment can activate some idle funds to play a "reservoir" effect, too much of the funds used for financial investment will crowd out the funds needed for the entity development of enterprises [2], inhibit the total factor productivity of enterprises' operational businesses, block enterprises' technological innovation [3], reduce enterprises' ability to bear risks [4], and causing the enterprise to detach from solid to empty. Therefore, the "crowding out" effect of the financialization of non-financial enterprises is greater than the "reservoir" effect.

Supply chain management focuses on coordination and cooperation among several business partners linked by material, capital and information flows [5]. The practice of supply chain management includes four parts: supply chain relationship, information sharing level, information quality level and delay [6]. Among them, customer relationship is the most basic content. Supply chain partners include suppliers of basic raw materials and parts, manufacturers, wholesalers, distributors, transporters, retailers, banks and financial institutions [7]. In this paper, it is generally summarized as upstream suppliers and downstream customers. At present, China's business environment has problems such as imperfect legal environment and insufficient market competition. The interconnection between supply chains is largely established through informal institutions, such as trust or the social network of enterprises. Moreover, Chinese listed companies generally rely on the supply chain, which is far more concentrated than other countries. For example, more than half of the major suppliers purchase more than 25%, and more than 45% of the major customer sales of enterprises exceed 25% among A-share listed companies in 2019, which indicates that there are huge differences in the cooperation scale of supply chain partners. The supply chain connection is actually a cooperative relationship of repeated game in the supply chain, so it will have an important impact on business decisions and economic consequences such as technological innovation [8], CSR performance of social responsibility [9], audit risk [10], cost structure, tax avoidance [11], bond credit spread [12] and business performance.

In a word, the risks and benefits of supply chain concentration will have an impact on the enterprise's financialization and relevant research focuses on two aspects. On the one hand, from the perspective of supply chain stability, the supply chain relationship emphasizes the industrial integration effect brought by the concentration of supply chain, which may hinder the enterprise financialization. Specifically, as the concentration of supply chain increases, supply and demand information will be better integrated into the production of enterprises, reducing agency problems [13]. On the other hand, from the perspective of the strength comparison of supply members, it is emphasized that the concentration of supply chain will make enterprises form strong dependence, which may aggravate the financialization of enterprises. Specifically, the concentration of supply chain will increase the business risk and cash flow risk, intensify the financing constraints of enterprises, inhibit enterprise innovation [7], increase the possibility of inefficient investment of enterprises [14] and ultimately damage the business performance of enterprises [15]. Because supply chain concentration is both "risk effect" and "revenue effect", this paper believes that supply chain concentration may not only aggravate the enterprise's financialization, but also hinder the enterprise's financialization. In order to determine the impact effect and further analyze the impact mechanism, this paper uses the relevant data of China's A-share listed companies from 2009 to 2021 to analyze the impact of supply chain concentration on enterprise financialization and substantiation.

This paper's knowledge fields mainly include supply chain management and enterprise financial investment, so this paper empirically analyzes the impact of supply chain concentration on enterprise financialization. It is noteworthy that many scholars have studied the economic consequences of customer concentration, but few have extended to suppliers and

supply chains. In addition, although many scholars have studied customer and enterprise financialization separately, few scholars have studied them together. And there are few studies on the specific impact mechanism of supply chain concentration. Compared with the existing research, this paper deepens the existing research from the internal characteristics of the enterprise to the external environment, promoting the research on customer concentration from the existing overall average effect to a more detailed level. There may be marginal contributions from the following aspects: First, Unlike Ak & Patatoukas [16] and Cao et al. [17], which only focuses on the impact of customer concentration on enterprise financialization, this paper integrates customer concentration and supplier concentration into the same analytical framework to investigate the impact on enterprise financialization and enrich relevant research on customer concentration. Second, the existing research on the mechanism of supply chain concentration affecting enterprise financialization mainly focuses on the enterprise's own behavior, just like the research of Huang et al. [11]. This paper introduces non-contractual upstream and downstream commercial credit as intermediary variables, and studies the specific impact mechanism which is of great significance for enriching the research on the mechanism of supply chain concentration on enterprise economic behavior. Third, from the perspective of supply chain concentration and enterprise financialization, this paper provides new evidence for mandatory disclosure of customer information from both risk and income aspects, and then provides reference for subsequent policy formulation.

## 2 Theoretical analysis and research hypothesis

This paper argues that the impact of supply chain concentration on enterprise financialization is shown in two aspects.

On the one hand, the "risk effect" brought by supply chain concentration will aggravate the enterprise's financialization and inhibit its substantiation. This impact is embodied in the following two items. Firstly, supply chain concentration will increase the business risk of enterprises [18]. Suppliers or customers choose to terminate cooperation suddenly or choose other enterprises, which will threaten the normal operation of the enterprise greatly. Moreover, enterprises are highly dependent on the supply chain, so they have to make concessions when negotiating, so as to increase the cost of purchasing production materials and reduce the cost of product sales. In order to maintain the relationship with main customers and suppliers, the enterprise will pay more entertainment and financial expenses [19], reducing the profit space of the enterprise. Supply chain concentration also forces enterprises to make proprietary investment, which leads to a fragile cost structure and increases the business risk from the perspective of cost. In order to avoid the operational risks brought by the concentration of supply chain, enterprises will try financial investment with high return rate and short return period, which is not conducive to the sustainable growth of enterprises [19]. Secondly, the concentration of supply chain will increase the capital risk of enterprises. In order to maintain a stable trading relationship with customers, enterprises often give more business credit to key customers. This kind of commercial credit is generally manifested in extending the term of accounts receivable and increasing the amount of accounts receivable. Thus, the increase of business credit reduces the asset turnover speed of enterprises, results in the fracture of the capital chain of enterprises, and increases the cash flow risk of enterprises. Furthermore, the concentration of supply chain may shorten the loan term, increase the bond spread [20], and increase the number of restrictive contracts [12]. As a result, enterprises face greater financing constraints, which may even lead to the collapse of stock prices [21]. In order to avoid abnormal cash flow, the enterprise will also tend to hold more cash or make some financial investments, which will aggravate the "crowding out effect" of enterprise financialization.

On the other hand, the "revenue effect" brought by supply chain concentration will hinder enterprise financialization and promote its substantiation. This impact is embodied in the following two items. Firstly, centralized supply chain will lead to the integration of supply and demand information. The mutual transmission of demand side and supply side information enables enterprises to obtain the surface needs of consumers, penetrate into the core needs of consumers, and tap the potential needs of consumers. It can also accelerate technological innovation from the supply side, adjust product parameters, improve product performance, and adapt to market demand. Then it can increase the main business income of enterprises, improve the core competitiveness of products, improve the operating efficiency of enterprises, reduce the risk of enterprises [17], promote the transformation and upgrading of enterprises. Secondly, supply chain concentration can give full play to the effect of industrial integration. The establishment of a stable relationship between enterprise and supply chain can realize the dual advantages of high asset specialization and low transaction costs. The high asset specificity of the enterprise can give play to the value creation effect of specialized assets, thereby expanding the market share of the enterprise and increasing the profit of the enterprise. The low transaction cost of an enterprise includes that the supply chain collaborative marketing reduces the enterprise's product sales cost, and the accurate placement of inventory improves the inventory management efficiency. The relational transaction also reduces the negotiation cost, and the flexible adjustment of production and operation improves the enterprise's operating efficiency. Furthermore, the integration of supply chain resources will bring a stable supply of goods and sellers. A good supply chain relationship can form a friendly external supervision mechanism for enterprises [22], thus affecting the investment duration and investment proportion of investors. In the long run, the concentration of supply chain relations can stabilize the sales revenue of enterprises, increase enterprise performance [23], affect the industrial governance structure and corporate governance level, promote the entity development of enterprises, and inhibit their financial development.

Based on the above analysis, this paper will propose two competitive hypotheses from the perspective of the impact of supply chain concentration on enterprise financialization:

H1: With other conditions unchanged, supply chain concentration will aggravate the enterprise's financialization and inhibit its substantiation.

H2: With other conditions unchanged, supply chain concentration hinder enterprise financialization and promote its substantiation.

## 3 Research design

### 3.1 Model design and variable definition

According to the research hypothesis, the model of this paper is as follows:

$$Fin_{it} = \alpha + \beta CC_{it} + \sum \alpha_k Controls_{it} + Year + Ind + \varepsilon_{it} \tag{1}$$

$$Fin_{it} = \alpha + \beta SC_{it} + \sum \alpha_k Controls_{it} + Year + Ind + \varepsilon_{it} \tag{2}$$

Among them, $i$ stands for enterprise, $t$ stands for year, $Fin_{it}$ represents the enterprise financialization of the explained variable. In order to further study, this paper introduces $Fix_{it}$ as a supplement to the study of financialization which is used to reflect the substantiation of enterprises. The core explanatory variables are customer concentration ($CC$) and supplier concentration ($SC$). $\beta$ is the estimated coefficient of the core explanatory variables. If $\beta > 0$, it is

indicated that supply chain concentration can aggravate enterprise financialization. The model also controls the fixed effects of industry (*Ind*) and year (*Year*).

Interpreted variable: Enterprise financialization. Referring to the practices of Xie et al. [24], Liu & Wei [3], this paper uses the proportion of financial assets in total assets to measure the level of enterprise financialization. Financial assets include monetary capital, net held to maturity investment, trading financial assets, derivative financial assets, net available for sale financial assets, net long-term equity investment, net dividend receivable and net interest receivable. In order to further verify the enterprise financialization, this paper introduces the second explained variable substantiation level from the perspective of "crowding out effect". This paper uses the proportion of assets used by enterprises for production innovation in total assets to measure the enterprise's substantiation level. Assets used for production innovation include net fixed assets, net amount of intangible assets and cash paid for acquisition and construction of fixed assets, intangible assets, and other long-term assets.

Core explanatory variable: Supply chain centralization. This paper analyzes from two perspectives: customer concentration and supplier concentration. Referring to the practices of Liu et al. [20] and Campello & Gao [12], this paper uses the proportion of the sum of sales of the top five customers in total sales (*CC*) as the measurement indicators of customer concentration. In the same way, the proportion of the sum of the purchase amount of the top five suppliers in the total purchase amount (*SC*) as the measurement indicators of supplier concentration.

Mediation variables: downstream business credit (*DTC*) and upstream business credit (*UTC*). In order to cope with the risks brought by supply chain and stabilize the supply chain relationship, enterprises must provide more commercial credit to customers, thus increasing the amount and duration of accounts receivable, intensifying the cash flow risk of enterprises, and intensifying the financial level of enterprises. This paper introduces the upstream and downstream commercial credit into the model respectively to study its impact on the concentration of the supply chain.

Based on the existing literature, this paper selects enterprise scale (*Size*), return on total assets (*ROA*), enterprise age (*Age*), asset liability ratio (*Lev*), long term debt ratio (*LD*), growth rate of operating income (*Growth*), fund turnover rate (*Turnover*), board size (*Bsize*), cash flow from operating activities (*Cfo*), proportion of independent directors (*Outdir*) as control variables. Cash holdings (*Cashhold*), proportion of accounts receivable (*RR*), KZ financing constraints (*KZ*) are the control variables added in the robustness test. In addition, this paper also sets dummy variables of year (*Year*) and industry (*Ind*) to control the impact of annual change trend. See Table 1 for specific variables.

## 3.2 Data description

The data in this paper are mainly from all A-share listed manufacturing enterprises from 2009 to 2021. The specific processing process is as follows: ST and delisting samples are removed; The data missing samples were eliminated and 17600 sample observations were finally obtained; The company's financial and governance data are derived from CSMAR and WIND databases; In order to eliminate the influence of abnormal values, all continuous variables were shrunk to 1% and 99% quantiles.

Table 2 reports the descriptive statistical results of variables. The average annual industrial investment (*Fix*) of A-share listed manufacturing companies is 0.281, and the maximum and minimum values are 0.679 and 0.0320 respectively, indicating that there is a large gap between the industrial investment levels of different enterprises. The average value of the degree of financialization (*Fin*) is 0.229, and the maximum and minimum values are 0.723 and 0.0280

**Table 1. Variable definition and meaning.**

| Variable Type | Variable name | Variable Symbol | Variable definition |
|---|---|---|---|
| Interpreted variable | Enterprise financialization | Fin | Financial assets/Total Assets |
| | Enterprise substantiation | Fix | Assets used for entity operation/Total Assets |
| Explanatory variable | Customer concentration | CC | Proportion of sales of top five customers in total sales |
| | Supplier concentration | SC | Proportion of purchase of top five suppliers in the total purchase |
| Intermediary variable | Business credit provided by enterprises to customers | DTC | (Accounts receivable + Notes receivable) / Total Assets |
| | Business credit provided by suppliers | UTC | (Notes payable + Accounts payable) / Total Assets |
| Control variable | Enterprise scale | Size | Ln (Total Assets) |
| | Return on total assets | ROA | Net profit / Total Assets |
| | Enterprise age | Age | Current year—registered year+1 |
| | Asset liability ratio | Lev | Total Assets |
| | Long term debt ratio | LD | Long term loans / Total Assets |
| | Cash flow from operating activities | Cfo | Net cash flow from operating activities / Total Assets |
| | Growth rate of operating income | Growth | (Operating income of the current year—operating income of the previous year) / operating income of the previous year |
| | Fund turnover rate | Turnover | Cash received from selling goods and providing services/Total Assets |
| | Board size | Board | Ln (number of directors+1) |
| | Proportion of independent directors | Outdir | Number of independent directors/number of directors |
| | Cash holdings | Cashhold | (Monetary capital + trading financial assets)/Total Assets |
| | Proportion of accounts receivable | RR | — |
| | KZ financing constraints | KZ | — |
| | particular year | Year | Dummy variable, thirteen years |
| | industry | Ind | Dummy variable, divided into 29 industries |

**Table 2. Descriptive statistical analysis of main variables.**

| Variables | Observations | Mean | Standard Deviation | Median | Min | Max |
|---|---|---|---|---|---|---|
| Fix | 17600 | 0.281 | 0.144 | 0.263 | 0.0320 | 0.679 |
| Fin | 17600 | 0.229 | 0.142 | 0.195 | 0.0280 | 0.723 |
| CC | 17600 | 0.319 | 0.208 | 0.266 | 0.0360 | 0.948 |
| SC | 17600 | 0.339 | 0.184 | 0.296 | 0.0690 | 0.913 |
| UTC | 17600 | 0.135 | 0.0930 | 0.114 | 0.00700 | 0.434 |
| DTC | 17600 | 0.164 | 0.107 | 0.149 | 0.00200 | 0.484 |
| Age | 17600 | 19.89 | 5.567 | 20 | 8 | 37 |
| Roa | 17600 | 0.0380 | 0.0680 | 0.0390 | -0.276 | 0.216 |
| Cfo | 17600 | 0.0500 | 0.0670 | 0.0480 | -0.146 | 0.250 |
| Size | 17600 | 22.05 | 1.152 | 21.90 | 19.37 | 25.44 |
| Lev | 17600 | 0.399 | 0.194 | 0.388 | 0.0520 | 0.904 |
| LD | 17600 | 0.0280 | 0.0480 | 0.00100 | 0 | 0.238 |
| Growth | 17600 | 0.172 | 0.363 | 0.115 | -0.497 | 2.158 |
| Turnover | 17600 | 0.601 | 0.372 | 0.516 | 0.0980 | 2.294 |
| Outdir | 17600 | 0.377 | 0.0540 | 0.357 | 0.333 | 0.571 |
| Board | 17600 | 2.107 | 0.187 | 2.197 | 1.609 | 2.565 |
| KZ | 17600 | 127.1 | 61.54 | 123.6 | 17.95 | 290.8 |
| Cashhold | 17600 | 4.087 | 7.073 | 2.677 | -18.66 | 48.84 |
| RR | 17600 | 77.07 | 25.16 | 85.90 | 0 | 100 |

**Table 3. Correlation analysis.**

|  | Fin | Fix | CC | SC | Roa | Lev | Growth |
|---|---|---|---|---|---|---|---|
| Fin | 1 | | | | | | |
| Fix | - | 1 | | | | | |
| CC | 0.036*** | -0.103*** | 1 | | | | |
| SC | 0.057*** | 0.036*** | 0.271*** | 1 | | | |
| Roa | 0.247*** | -0.129*** | -0.072*** | -0.020*** | 1 | | |
| Lev | -0.372*** | 0.162*** | -0.037*** | -0.121*** | -0.393*** | 1 | |
| Growth | -0.024*** | -0.060*** | 0.068*** | 0.034*** | 0.257*** | 0.025*** | 1 |
| Turnover | -0.022*** | 0.033*** | -0.146*** | 0.022*** | 0.146*** | 0.142*** | 0.037*** |

respectively, indicating that there is a large gap between the financial investment levels. From the perspective of comparison between industrial investment and financial investment, although the average and median of industrial investment are higher than financial investment, the average of financial investment accounts for more than 20% of the total investment, which means that the overall level of financialization of Chinese enterprises is high. The average value of CC customer concentration is 0.319, the maximum value is 0.948, and the minimum value is 0.0360; The average value of SC supplier concentration is 0.339, the minimum value is 0.0690, and the maximum value is 0.913. This shows that different manufacturing enterprises have significant differences in customer and supplier concentration. As manufacturing enterprises have a higher degree of supply chain concentration than commercial, financial, and public utilities, the measurement results of supply chain concentration in this paper are slightly higher than others [25]. Control variables are basically consistent with existing research.

This paper analyzes the correlation of the main variables in the model. According to the analysis in Table 3, it is found that customer concentration (*CC*) is positively correlated with enterprise financialization (*Fin*) and negatively correlated with enterprise substantiation (*Fix*), which is significant at the level of 1%. The concentration of suppliers (*SC*) is positively correlated with the financialization (*Fin*) and substantiation (*Fix*) of enterprises, and it is significant at the level of 1%. The above results need further empirical testing.

## 4 Empirical analyses

### 4.1 Benchmark regression

According to hypothesis H1 and H2 and model (1) and (2), regression analysis is conducted for customer concentration and supplier concentration respectively. The regression results are shown in Table 4. The four columns of regression results control the double fixed effects of year and industry, and control variables are introduced.

Columns (1)—(4) in Table 4 report the impact of supply chain concentration on the financialization and substantiation of enterprises. For financialization (*Fin*), the estimated coefficient of *CC* is 0.040, and *SC is* 0.054. They are significant at the level of 1%. For substantiation (*Fix*), the estimated coefficient of *CC* is -0.069, and *SC is* -0.064. They are also significant at the level of 1%. From the above analysis, both supplier and customer concentration will inhibit the enterprises substantiation and aggravate their financialization. The above conclusions prove the hypothesis H1.

### 4.2 Robustness test

Considering that there may be a reverse causal relationship between supply chain concentration and enterprise financialization, this paper uses instrumental variable method to conduct

Table 4. Benchmark regression results.

| Variable | Customer concentration | | Supplier concentration | |
|---|---|---|---|---|
| | *Fin* | *Fix* | *Fin* | *Fix* |
| | (1) | (2) | (3) | (4) |
| CC | 0.040*** | -0.069*** | | |
| | (2.95) | (-4.81) | | |
| SC | | | 0.054*** | -0.064*** |
| | | | (4.91) | (-5.41) |
| Age | 0.005 | -0.005 | 0.005 | -0.005 |
| | (1.09) | (-1.39) | (1.06) | (-1.37) |
| Roa | 0.066*** | -0.196*** | 0.066*** | -0.197*** |
| | (3.29) | (-9.52) | (3.28) | (-9.68) |
| LD | -0.184*** | 0.196*** | -0.185*** | 0.198*** |
| | (-6.38) | (5.54) | (-6.42) | (5.58) |
| Cfo | 0.214*** | 0.117*** | 0.215*** | 0.115*** |
| | (11.79) | (7.42) | (11.84) | (7.29) |
| Size | 0.011*** | -0.025*** | 0.012*** | -0.025*** |
| | (2.58) | (-5.03) | (2.79) | (-5.07) |
| Lev | -0.200*** | 0.047*** | -0.200*** | 0.047*** |
| | (-11.49) | (2.66) | (-11.56) | (2.65) |
| Growth | -0.006** | -0.006** | -0.006** | -0.006** |
| | (-2.14) | (-2.02) | (-2.13) | (-2.17) |
| Turnover | -0.042*** | 0.003 | -0.043*** | 0.004 |
| | (-4.69) | (0.32) | (-4.84) | (0.56) |
| Outdir | -0.032 | 0.001 | -0.033 | 0.005 |
| | (-0.86) | (0.03) | (-0.91) | (0.14) |
| Board | -0.007 | -0.012 | -0.008 | -0.010 |
| | (-0.57) | (-1.05) | (-0.65) | (-0.93) |
| Constant | 0.068 | 0.909*** | 0.046 | 0.909*** |
| | (0.61) | (8.00) | (0.41) | (8.02) |
| N | 17,600 | 17,600 | 17,600 | 17,600 |
| Ind FE | YES | YES | YES | YES |
| Year FE | YES | YES | YES | YES |
| Adj-R² | 0.128 | 0.0852 | 0.129 | 0.0846 |

*Notes*: The standard error of robustness is shown in brackets;

*, ** and *** are significant at 10%, 5% and 1% levels respectively, the same below; Clustering adjustment has been made at the company level, the same below.

endogenous test. The paper selects the average customer information disclosure level of other enterprises in the same province and the same industry in a lag period (*LMCC*), as the instrument variable of customer concentration; Similarly, the average disclosure level of supplier information (*LMSC*) of other enterprises in the same province and industry with a lag of one period is used as the instrument variable of supplier concentration. The 2SLS method is used for estimation. The regression results are shown in columns (1)-(4) of Table 5. The statistical values of Kleibergen-Paap rk LM and Kleibergen-Paap rk Wald F in columns (2) and (4) are far greater than their respective critical values, and have passed the tests of unrecognizability and weak instrumental variables. In the first stage, both LMCC and LMSC are significantly positive at the level of 1%, indicating that instrumental variables are strongly correlated with explanatory variables. In the second stage, the promotion of customer concentration (*CC*) and supplier

**Table 5. Endogenous test results.**

| Variables | Customer concentration | | Supplier concentration | |
|---|---|---|---|---|
| | CC | FIN | SC | FIN |
| | (1) | (2) | (3) | (4) |
| LMCC | 0.881*** | | | |
| | (55.16) | | | |
| CC | | 0.072*** | | |
| | | (4.33) | | |
| LMSC | | | 0.842*** | |
| | | | (47.76) | |
| SC | | | | 0.085*** |
| | | | | (4.07) |
| N | 14,365 | 14,365 | 14,365 | 14,365 |
| Controls | Yes | Yes | Yes | Yes |
| Ind FE | Yes | Yes | Yes | Yes |
| Year FE | Yes | Yes | Yes | Yes |
| Adj-$R^2$ | | 0.193 | | 0.195 |
| Kleibergen-Paap rk LM | | 957.118 | | 761.131 |
| Kleibergen-Paap rk Wald F | | 3042.511 | | 2281.477 |

concentration (*SC*) on enterprise financialization (*Fin*) is significant at the level of 1%, indicating that the conclusion of this study is still reliable after controlling endogenous problems.

In addition to the endogenous test, this paper also carried out three aspects of robustness tests. First, replace the core explanatory variables. Change the top five customers or suppliers to the top one customers and suppliers. Select the proportion of the sales volume of the largest customer in the total sales volume (*BCC*) and the proportion of the purchase volume of the largest supplier in the total purchase volume (*BSC*) as the new core explanatory variables. The regression results are as described in column (1) and (2) of Table 6. The first largest customer concentration and the first largest supplier concentration have a significant promotion effect on financialization (*Fin*) at the level of 1%, and a significant inhibition effect on substantiation (*Fix*) at the level of 1%, which confirms the previous conclusion.

Second, add control variables. In order to confirm the above research, this paper adds three control variables: cash holdings (*Cashhold*), accounts receivable ratio (*RR*) and KZ financing constraints (*KZ*), which are all from the Wind database. The regression results are as described in columns (3) and (4) of Table 6. The promotion of *CC* and *SC* on financialization (*Fin*) is significant at the level of 1%, and the inhibition on substantiation (*Fix*) is significant at the level of 1%, indicating that the conclusions above are still valid.

Third, reselect the sample. The centralized information of China's supply chain has been completely disclosed since 2012. The data disclosed before 2012 is incomplete, which may have an impact on the regression results. Therefore, the data from 2009 to 2012 will be deleted and the regression will be conducted again. The regression results are as described in columns (5) and (6) of Table 6. The promotion of *CC* and *SC* on financialization (*Fin*) is significant at the level of 1%, and the inhibition on substantiation (*Fix*) is significant at the level of 1%, indicating that the results of this paper are relatively stable.

## 4.3 Mechanism analysis

The previous research shows that both customer concentration and supplier concentration can significantly promote the financialization of enterprises and inhibit their substantiation.

**Table 6. Robustness test.**

| Variables | Replace the core explanatory variables | | Add control variables | | Reselect the sample | |
|---|---|---|---|---|---|---|
| | *Fin* | *Fix* | *Fin* | *Fix* | *Fin* | *Fix* |
| | (1) | (2) | (3) | (4) | (5) | (6) |
| Customer concentration | | | | | | |
| BCC | 0.037** | -0.062*** | | | | |
| | (2.26) | (-3.31) | | | | |
| CC | | | 0.038*** | -0.068*** | 0.036*** | -0.072*** |
| | | | (2.89) | (-4.78) | (2.68) | (-5.00) |
| Constant | -0.021 | 1.052*** | 0.166 | 0.877*** | 0.030 | 0.952*** |
| | (-0.17) | (8.22) | (1.49) | (7.82) | (0.24) | (7.69) |
| N | 13,001 | 13,001 | 17,600 | 17,600 | 16,970 | 16,970 |
| Controls | Yes | Yes | Yes | Yes | Yes | Yes |
| Ind FE | Yes | Yes | Yes | Yes | Yes | Yes |
| Year FE | Yes | Yes | Yes | Yes | Yes | Yes |
| Adj-R2 | 0.130 | 0.0938 | 0.130 | 0.0858 | 0.103 | 0.0851 |
| Supplier concentration | | | | | | |
| BSC | 0.046*** | -0.068*** | | | | |
| | (2.86) | (-3.41) | | | | |
| SC | | | 0.053*** | -0.064*** | 0.048*** | -0.066*** |
| | | | (4.87) | (-5.38) | (4.23) | (-5.55) |
| Constant | -0.000 | 1.031*** | 0.144 | 0.876*** | 0.013 | 0.950*** |
| | (-0.00) | (8.00) | (1.30) | (7.84) | (0.10) | (7.74) |
| N | 12,844 | 12,844 | 17,600 | 17,600 | 16,970 | 16,970 |
| Controls | Yes | Yes | Yes | Yes | Yes | Yes |
| Ind FE | YES | YES | YES | YES | YES | YES |
| Year FE | YES | YES | YES | YES | YES | YES |
| Adj-R² | 0.132 | 0.0931 | 0.132 | 0.0852 | 0.104 | 0.0842 |

The conclusion is still valid after controlling the endogenous problem. Next, we will further study the specific mechanism of customer concentration and supplier concentration to promote the enterprise's financialization.

According to the existing research, the dependence of enterprises on the supply chain will bring certain business risks and cash flow risks to enterprises. In order to cope with the risks brought by the concentration of the supply chain and stabilize the supply chain relationship, enterprises must provide more commercial credit to customers, thus increasing the amount and accounting period of accounts receivable, reducing the capital turnover rate of enterprises, and intensifying the cash flow risk of enterprises. Therefore, in order to avoid risks, enterprises will tend to choose financial investments with short duration and high return on investment, which will aggravate the financial level of manufacturing enterprises. Therefore, it is feasible to introduce commercial credit into the model as an intermediary variable. This paper verifies the impact of commercial credit as an intermediary variable on the financialization of manufacturing enterprises from two paths of customer and supplier. The specific results are shown in Table 7.

First, the intermediary effect of business credit provided by enterprises to customers. The columns (1)—(3) in Table 7 are the regression results of the path "customer concentration → business credit provided by enterprises → enterprise financialization". The coefficient of *CC* in Column (1) has a positive and significant level of 1% of business credit (*DTC*), indicating that

**Table 7. Mechanism analysis results.**

| Variables | DTC | Fin | Fix | UTC | Fin | Fix |
|---|---|---|---|---|---|---|
| | (1) | (2) | (3) | (4) | (5) | (6) |
| CC | 0.022*** | 0.049*** | -0.067*** | | | |
| | (2.59) | (3.74) | (-4.72) | | | |
| DTC | | -0.401*** | -0.121*** | | | |
| | | (-17.08) | (-4.77) | | | |
| SC | | | | 0.008 | 0.055*** | -0.064*** |
| | | | | (1.43) | (4.98) | (-5.36) |
| UTC | | | | | -0.115*** | -0.081** |
| | | | | | (-3.82) | (-2.38) |
| Constant | 0.220** | 0.156 | 0.935*** | -0.090 | 0.035 | 0.901*** |
| | (2.47) | (1.50) | (8.21) | (-1.43) | (0.32) | (7.96) |
| N | 17,600 | 17,600 | 17,600 | 17,600 | 17,600 | 17,600 |
| Controls | Yes | Yes | Yes | Yes | Yes | Yes |
| Ind FE | YES | YES | YES | YES | YES | YES |
| Year FE | YES | YES | YES | YES | YES | YES |
| Adj-$R^2$ | 0.124 | 0.175 | 0.0917 | 0.217 | 0.131 | 0.0863 |

customer concentration has significantly increased the business credit provided by enterprises to key customers. From the results of columns (2) and (3), the coefficient of the influence of customer concentration (CC) on financialization (Fin) is significantly positive at the level of 1%, and the coefficient of the influence on substantiation (Fix) is significantly negative at the level of 1%, indicating that customer concentration promotes corporate financialization by increasing downstream business credit, while inhibiting its substantiation. It shows that downstream commercial credit plays a part of intermediary effect in the relationship between customer concentration, enterprise financialization and substantiation.

Second, the intermediary effect of the commercial credit provided by suppliers. The columns (4)—(6) in Table 7 are the regression results of the path "supplier concentration → commercial credit provided by suppliers → enterprise financialization". The supplier concentration (SC) in column (4) is not significant, indicating that the supplier concentration will not significantly increase the commercial credit provided by suppliers. From the results of columns (5) and (6), the coefficient of the influence of supplier concentration (SC) on enterprise financialization and substantiation are significantly at the level of 1%, indicating that customer concentration promotes enterprise financialization and inhibits its substantiation. However, upstream commercial credit has not played an intermediary role in the relationship between supplier concentration and enterprise financialization.

## 5 Extensive research

The previous article confirmed that supply chain concentration has a significant role in promoting enterprise financialization and inhibiting its materialization, and the commercial credit provided by downstream plays a part of the intermediary role. The concern is that the above results may be different in different groups. The reasons are as follows: on the one hand, there are serious property right discrimination and scale discrimination in China's financial market, and the concentration of supply chain will have different effects on the financialization of enterprises with different property rights and different scales; On the other hand, there are significant differences between different enterprises in terms of industry competition intensity

and regional development level, which will also have an impact on the above empirical results. Based on this, this paper conducts extensive research from four aspects: the nature of property rights, the size of enterprises, the intensity of industry competition, and the level of regional economic development where enterprises are located.

## 5.1 Heterogeneity analysis of the property rights nature

Since the investment decision of state-owned enterprises is to some extent the embodiment of the national will, and it is inevitable to be adjusted by the national policy while pursuing enterprise profits, the purchase and sales behavior of state-owned manufacturing industry is essentially a special behavior of policy adjustment and market transaction. According to the property rights of enterprises, this paper uses the method of introducing dummy variables to divide the samples: state-owned enterprises are recorded as 1, and non-state-owned enterprises are recorded as 0. The grouped regression results according to the property right nature are shown in Table 8.

In column (1) of Table 8, the regression coefficients of customer concentration (CC) in state-owned enterprises and non-state-owned enterprises are 0.042 and 0.032 respectively,

**Table 8. Heterogeneity analysis of the property right nature.**

| Variables | Customer concentration | | | Supplier concentration | | |
|---|---|---|---|---|---|---|
| | Fin | DTC | Fin | Fin | UTC | Fin |
| | (1) | (2) | (3) | (4) | (5) | (6) |
| SOE = 1 | | | | | | |
| CC | 0.042* | 0.028* | 0.052** | | | |
| | (1.85) | (1.76) | (2.37) | | | |
| DTC | | | -0.364*** | | | |
| | | | (-9.23) | | | |
| SC | | | | 0.057*** | -0.001 | 0.057*** |
| | | | | (2.94) | (-0.11) | (2.93) |
| UTC | | | | | | -0.072 |
| | | | | | | (-1.57) |
| N | 5,854 | 5,854 | 5,854 | 5,854 | 5,854 | 5,854 |
| Controls | Yes | Yes | Yes | Yes | Yes | Yes |
| Ind FE | YES | YES | YES | YES | YES | YES |
| Year FE | YES | YES | YES | YES | YES | YES |
| Adj-R² | 0.0932 | 0.117 | 0.144 | 0.0954 | 0.206 | 0.0965 |
| SOE = 0 | | | | | | |
| CC | 0.032* | 0.017 | 0.038** | | | |
| | (1.90) | (1.63) | (2.38) | | | |
| DTC | | | -0.393*** | | | |
| | | | (-15.04) | | | |
| SC | | | | 0.038*** | 0.014* | 0.041*** |
| | | | | (2.90) | (1.92) | (3.08) |
| UTC | | | | | | -0.153*** |
| | | | | | | (-3.93) |
| N | 11,116 | 11,116 | 11,116 | 11,116 | 11,116 | 11,116 |
| Controls | Yes | Yes | Yes | Yes | Yes | Yes |
| Ind FE | YES | YES | YES | YES | YES | YES |
| Year FE | YES | YES | YES | YES | YES | YES |
| Adj-R² | 0.115 | 0.131 | 0.158 | 0.116 | 0.217 | 0.120 |

which are significant at the level of 10%. In column (4), the regression coefficient of supplier concentration (*SC*) in state-owned enterprises and non-state-owned enterprises is 0.057 and 0.038 respectively, which is significant at the level of 1%. So, there is a significant difference between the two groups. This shows that compared with non-state-owned enterprises, supply chain concentration plays a stronger role in promoting the financialization of state-owned enterprises. This may be because state-owned enterprises are subject to the dual restrictions of the market and the government in the operation process, have a strong dependence on the supply chain, and have less choice in response to operational risks and cash flow risks, leading to their relatively high level of financialization.

In addition, the intermediary effect regression results of columns (2) and (3) of state-owned enterprises show that the regression coefficient of customer concentration (*CC*) on the business credit (*DTC*) provided by enterprises to customers is 0.028, which is significant at the level of 10%. That means the downstream business credit has some intermediary effects on the financialization of state-owned enterprises. The results of columns (2) and (3) of non-state-owned enterprises show that the downstream business (*DTC*) credit has no intermediary effect in the process of non-state-owned enterprises financialization, but the regression coefficient of upstream business credit (*UTC*) in columns (4) and (5) is 0.014, which is significant at the level of 10%, which means that the business credit provided by enterprises to customers has some intermediary effect in the impact of state-owned enterprises' financialization. The enterprise's suppliers focus on improving the enterprise's financial level by increasing the upstream business credit. In short, state-owned enterprises must provide more downstream business credit to maintain cooperation with key customers, while non-state-owned enterprises have accepted more upstream business credit to increase their financial level.

## 5.2 Heterogeneity analysis of the size of enterprises

The size of an enterprise reflects its competitiveness to a certain extent. Large enterprises have larger market shares and more stable supply chain relationships. According to the size of the enterprise, this paper uses the median method to divide the samples: the samples above the median of the enterprise scale as 1, and those below the median of the enterprise scale as 0. The grouping regression results by size are shown in Table 9.

In Column (1) of Table 9, the regression coefficient of customer concentration (*CC*) is 0.034 for large enterprises and 0.050 for small enterprises respectively. The former is significant at the level of 10%, while the latter is significant at the level of 5%. There is a significant difference between the two groups. In column (4), the regression coefficient of supplier concentration (*SC*) in large enterprises and small enterprises are 0.044, both of which are significant at the level of 1%, indicating that there is no difference between the two groups in the impact of supplier concentration on enterprise financialization. This shows that compared with large enterprises, customer concentration has a stronger role in promoting the financialization of small enterprises, while supplier concentration has no obvious difference. This may be because small enterprises have a weak ability to avoid risks in the face of operating risks and cash flow risks brought by customers. And they have fewer coping measures. Small enterprises face fierce market competition and are more likely to be eliminated from the market. Therefore, small enterprises are eager to invest in financial assets more easily and intensify the financialization of enterprises.

In addition, the intermediary effect regression results of columns (2)—(3) and (5)—(6) of large enterprises show that the regression coefficient of customer concentration (*CC*) on downstream business credit (*DTC*) is 0.011, which has not passed the significance test, indicating that the business credit provided by enterprises to customers has no intermediary effect on

**Table 9. Heterogeneity analysis of the enterprise size.**

| Variable | Customer concentration | | | Supplier concentration | | |
|---|---|---|---|---|---|---|
| | *Fin* | *DTC* | *Fin* | *Fin* | *UTC* | *Fin* |
| | (1) | (2) | (3) | (4) | (5) | (6) |
| *Size = 1* | | | | | | |
| CC | 0.034* | 0.011 | 0.038** | | | |
| | (1.87) | (1.00) | (2.14) | | | |
| DTC | | | -0.357*** | | | |
| | | | (-12.64) | | | |
| SC | | | | 0.044*** | 0.016** | 0.046*** |
| | | | | (2.85) | (2.11) | (2.97) |
| UTC | | | | | | -0.112*** |
| | | | | | | (-2.93) |
| N | 8,603 | 8,603 | 8,603 | 8,603 | 8,603 | 8,603 |
| Controls | Yes | Yes | Yes | Yes | Yes | Yes |
| Ind FE | YES | YES | YES | YES | YES | YES |
| Year FE | YES | YES | YES | YES | YES | YES |
| Adj-$R^2$ | 0.0814 | 0.167 | 0.127 | 0.0827 | 0.192 | 0.0855 |
| *Size = 0* | | | | | | |
| CC | 0.050** | 0.028** | 0.062*** | | | |
| | (2.49) | (2.06) | (3.25) | | | |
| DTC | | | -0.445*** | | | |
| | | | (-13.15) | | | |
| SC | | | | 0.044*** | -0.000 | 0.044*** |
| | | | | (2.72) | (-0.05) | (2.72) |
| UTC | | | | | | -0.189*** |
| | | | | | | (-3.91) |
| N | 8,367 | 8,367 | 8,367 | 8,367 | 8,367 | 8,367 |
| Controls | Yes | Yes | Yes | Yes | Yes | Yes |
| Ind FE | YES | YES | YES | YES | YES | YES |
| Year FE | YES | YES | YES | YES | YES | YES |
| Adj-$R^2$ | 0.117 | 0.0988 | 0.164 | 0.116 | 0.182 | 0.121 |

state-owned enterprise financialization. But the regression coefficient of business credit (*UTC*) provided by suppliers to enterprises is 0.016. It is established at the significance level of 5%, and the influence of supplier concentration (*SC*) on enterprise financialization (*Fin*) rises from 0.044 to 0.046 after the intermediary variable added, which indicates that the supplier concentration of large enterprises improves the enterprise financialization level by increasing upstream business credit. The downstream commercial credit of the small business group also has a similar intermediary effect, while the upstream commercial credit does not.

## 5.3 Heterogeneity analysis of the industry competition

Enterprises with fierce industry competition will face higher business risk and cash flow risk, but they will also become the driving force for innovation and transformation of enterprises. Therefore, grouping according to the intensity of industry competition is of great significance for the impact of supply chain concentration. This paper uses "*Regind*" to measure the industry competition. In combination with the situation of the manufacturing industry itself, according to the industry classification name of the CSRC in 2012, the industries C25, C31, C32, C36 and

**Table 10. Heterogeneity analysis of the industry competition.**

| Variable | Customer concentration | | | Supplier concentration | | |
|---|---|---|---|---|---|---|
| | *Fin* | *DTC* | *Fin* | *Fin* | *UTC* | *Fin* |
| | (1) | (2) | (3) | (4) | (5) | (6) |
| Regind = 1 | | | | | | |
| CC | 0.060* | 0.036 | 0.075** | | | |
| | (1.87) | (1.61) | (2.56) | | | |
| DTC | | | -0.409*** | | | |
| | | | (-7.29) | | | |
| SC | | | | 0.038 | 0.012 | 0.039 |
| | | | | (1.56) | (0.90) | (1.61) |
| UTC | | | | | | -0.137* |
| | | | | | | (-1.93) |
| N | 2,351 | 2,351 | 2,351 | 2,351 | 2,351 | 2,351 |
| Controls | YES | YES | YES | YES | YES | YES |
| Ind FE | YES | YES | YES | YES | YES | YES |
| Year FE | YES | YES | YES | YES | YES | YES |
| Adj-R² | 0.118 | 0.113 | 0.189 | 0.115 | 0.219 | 0.119 |
| Regind = 0 | | | | | | |
| CC | 0.036** | 0.020** | 0.044*** | | | |
| | (2.46) | (2.14) | (3.05) | | | |
| DTC | | | -0.400*** | | | |
| | | | (-15.26) | | | |
| SC | | | | 0.058*** | 0.007 | 0.058*** |
| | | | | (4.70) | (1.08) | (4.75) |
| UTC | | | | | | -0.109*** |
| | | | | | | (-3.24) |
| N | 15,249 | 15,249 | 15,249 | 15,249 | 15,249 | 15,249 |
| Controls | YES | YES | YES | YES | YES | YES |
| Ind FE | YES | YES | YES | YES | YES | YES |
| Year FE | YES | YES | YES | YES | YES | YES |
| Adj-R² | 0.105 | 0.129 | 0.147 | 0.107 | 0.213 | 0.109 |

C37 in the manufacturing industry number are defined as regulated industries. Those industries that belong to these industries are defined as 1, and those that do not belong to them are defined as 0. *Regind* = 1 means the industry is highly competitive, while *Regind* = 0 means the industry is not highly competitive. The results of group inspection are shown in Table 10.

In the column (1) of Table 10, the regression coefficient of customer concentration (*CC*) is 0.060 for enterprises with fierce industry competition and 0.036 for enterprises with weak industry competition respectively. The former is significant at the level of 10%, while the latter is significant at the level of 5%. This shows that customer concentration has a greater impact on the financialization of enterprises with fierce industry competition. In column (4), the regression coefficient of SC is 0.038 for enterprises with fierce industry competition and 0.058 for enterprises with weak industry competition. The former is not significant, while the latter is significant at the level of 1%. This shows that the concentration of suppliers only has a significant impact on the financialization of enterprises with low industrial competition. This may be because when the industry is highly competitive, enterprises face greater pressure to survive, which makes enterprises must rely on suppliers to stabilize their sales revenue, while

enterprises with less intense industry competition are more affected by the concentration of upstream suppliers.

The regression coefficient of the CC in column (2) of enterprises with weak industrial competition to the downstream commercial credit is 0.02, which is significant at the 5% significance level, indicating that the downstream commercial credit has played a part of the intermediary role in the process of the impact of customer concentration on the enterprise financialization. And according to the regression results in column (3), it can be found that customer concentration can further aggravate the enterprise financialization by promoting the enterprise's commercial credit. The upstream supplier side has no similar intermediary effect.

### 5.4 Heterogeneity analysis of the regional economic development

Different regional environments have different impacts on enterprise financialization. The motivation of enterprises to choose financialization not only depends on the survival needs of enterprises, but also involves the investment needs. How does the regional economic development affect the concentration of supply chain on the level of enterprise financialization? Is there any difference in the impact between the east and the midwestern regions? In this paper, the registered area East is used to measure the economic development of the region where the enterprise is located. If the enterprise is in Beijing, Tianjin, Hebei, Shanghai, Jiangsu, Zhejiang, Fujian, Shandong, Guangdong and Hainan, the value is 1, otherwise it is 0. The results of group inspection are shown in Table 11.

In Column (1) of Table 11, the regression coefficient of CC is 0.001 for enterprises in the east and 0.040 for enterprises in the midwestern regions. The former is not significant, while the latter is significant at the level of 1%. In column (4), the regression coefficient of SC is 0.054 for enterprises in the east and 0.046 for enterprises in the midwestern regions respectively. The former is not significant, while the latter is significant at the level of 1%. This shows that supply chain concentration only has a significant impact on the midwestern enterprises with poor economic development. This may be because the information technology level of enterprises in the eastern region is relatively developed, and the motivation of enterprise innovation is sufficient. Hence enterprises will choose digital transformation technology innovation to enhance their core competitiveness, reduce the motivation of investing in financial assets and inhibit their financialization. Enterprises in the central and western regions face greater information asymmetry and corporate risk, which will aggravate their motivation for financialization.

The intermediary effect regression results of columns (2)—(3) and (5)—(6) of enterprises in the midwestern regions show that the regression coefficient of CC to downstream business credit *(DTC)* is 0.024, which is significant at the level of 1%. This shows that the downstream commercial credit has some intermediary effect on the enterprise financialization in the midwestern regions, and customers concentration increasing the downstream commercial credit to further intensify the enterprise financialization level. The upstream commercial credit has no similar intermediary effect.

## 6 Conclusions and policy implications

In recent years, with the decreasing dividend of enterprises, many manufacturing industries have chosen financial investment. The crowding out effect brought by enterprise financialization will damage the industrial development of enterprises, inhibit their technological innovation, and have a certain impact on the transformation and upgrading of China's manufacturing industry and high-quality economic development. This paper theoretically analyzes the impact of supply chain concentration on enterprise financialization and

**Table 11. Heterogeneity analysis of the regional economic development.**

| Variable | Customer concentration | | | Supplier concentration | | |
|---|---|---|---|---|---|---|
| | *Fin* | *DTC* | *Fin* | *Fin* | *UTC* | *Fin* |
| | (1) | (2) | (3) | (4) | (5) | (6) |
| *East = 1* | | | | | | |
| CC | 0.001 | 0.017 | 0.008 | | | |
| | (0.03) | (0.57) | (0.20) | | | |
| DTC | | | -0.382*** | | | |
| | | | (-5.47) | | | |
| SC | | | | 0.054 | 0.016 | 0.054 |
| | | | | (1.51) | (1.21) | (1.53) |
| UTC | | | | | | -0.026 |
| | | | | | | (-0.27) |
| N | 2,186 | 2,186 | 2,186 | 2,186 | 2,186 | 2,186 |
| Controls | YES | YES | YES | YES | YES | YES |
| Ind FE | YES | YES | YES | YES | YES | YES |
| Year FE | YES | YES | YES | YES | YES | YES |
| Adj-R² | 0.0738 | 0.138 | 0.113 | 0.0764 | 0.286 | 0.0760 |
| *East = 0* | | | | | | |
| CC | 0.040*** | 0.024*** | 0.049*** | | | |
| | (2.85) | (2.58) | (3.59) | | | |
| DTC | | | -0.379*** | | | |
| | | | (-16.11) | | | |
| SC | | | | 0.046*** | 0.007 | 0.047*** |
| | | | | (3.96) | (1.18) | (4.04) |
| UTC | | | | | | -0.132*** |
| | | | | | | (-4.29) |
| N | 15414 | 15414 | 15414 | 15414 | 15414 | 15414 |
| Controls | YES | YES | YES | YES | YES | YES |
| Ind FE | YES | YES | YES | YES | YES | YES |
| Year FE | YES | YES | YES | YES | YES | YES |
| Adj-R² | 0.111 | 0.123 | 0.156 | 0.112 | 0.203 | 0.115 |

substantiation and its mechanism, and selects the relevant data of A-share listed companies in China from 2009 to 2020 for empirical testing. The results show that: First, the concentration of supply chain will significantly aggravate the financialization of enterprises and this core conclusion is still valid after robustness and endogenous treatment. Therefore, the government needs to further disclose the supply chain relationship information of enterprises, coordinate the competitiveness between supply chains, and promote the integration of supply chain resources. In addition, the downstream commercial credit plays a very good role in mediating. Therefore, the government should pay more attention to the commercial credit to ensure the normal flow of enterprise funds. Finally, through the heterogeneity analysis, the impact of supply chain concentration on enterprises financialization is significantly different in state-owned enterprises and minor enterprises, and more significant in enterprises with less competitive industries and located in the midwestern regions. Therefore, the government needs to create a good business environment and support the development of minor enterprises and enterprises in the midwestern regions. Although this paper analyzes the impact of supply chain concentration on enterprise financialization, it ignores the role of supply chain activities and supply

chain efficiency in supply chain management. Therefore, future research should start from a deeper and more comprehensive supply chain management, and study the impact of supply chain management risks and benefits on enterprise economic behavior. In addition, this study can also promote relevant research in other countries.

Based on the above research results, this paper puts forward the following policy recommendations: First, as the concentration of supply chain aggravates the financialization of enterprises, the government needs regulate the profit-seeking behavior and promote the construction of positive cooperative relationship between upstream and downstream enterprises in the supply chain. With regard to whether the CSRC has required the disclosure of supply chain information, this paper believes that regulators should not only disclose the sales information of the top five customers, but also require further disclosure of the basic characteristics of upstream and downstream enterprises, so as to help information users more objectively evaluate the business status of enterprises. Moreover, according to the analysis of commercial credit as an intermediary variable mechanism, the government should pay attention to the important role of enterprise commercial credit as an informal financing channel, improve the relevant laws and regulations on debt governance, and protect the interests of creditors from both legal and moral aspects. Finally, according to the results of the heterogeneity analysis, the government should actively provide a good business environment for enterprises, give certain policy support to enterprises with fierce industrial competition and enterprises in the midwestern regions, and support the development of minor enterprises.

There are also some limitations in this paper: First, this paper only studies the samples of listed companies, but the research of non-listed companies is not involved in this paper. So, the conclusion of this paper is only applicable to listed companies. Future research can be expanded according to the unlisted companies. Second, there are still some problems in the endogenous treatment of this paper. Compared with the enterprise financialization, the customer concentration is not an exogenous variable, so it is difficult to find an instrumental variable that does not affect the enterprise financialization at all. Future research will try to find out the policy impact that affects the concentration of corporate customers, and carry out research with the method of policy identification.

## Supporting information

**S1 File.**
(ZIP)

## Author Contributions

**Conceptualization:** Huanhuan He, Zongwen Zuo.

**Data curation:** Huanhuan He, Zongwen Zuo.

**Funding acquisition:** Zongwen Zuo.

**Methodology:** Huanhuan He.

**Software:** Huanhuan He.

**Supervision:** Zongwen Zuo.

**Writing – original draft:** Huanhuan He.

**Writing – review & editing:** Zongwen Zuo.

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
