## [Decision Letter · Decision Letter 0]

8 Dec 2022

PONE-D-22-30905Supply Chain Concentration and Enterprise Financialization: Evidence from Listed Companies in China's Manufacturing IndustryPLOS ONE

Dear Dr. Zuo,

Thank you for submitting your manuscript to PLOS ONE. After careful consideration, we feel that it has merit but does not fully meet PLOS ONE’s publication criteria as it currently stands. Therefore, we invite you to submit a revised version of the manuscript that addresses the points raised during the review process.

ACADEMIC EDITOR:

1. Both reviewers raised questions about the introduction and contribution. Please take them seriously.

2. The model obviously has endogenous problems. It is suggested to add endogenous discussion in the robustness test.

3.There are a lot of grammatical errors.Should read the manuscript carefully to eliminate those errors and, if possible, have a native speakers to proofread the paper.

We look forward to receiving your revised manuscript.

Kind regards,

Enji Li, ph.D.

Academic Editor

PLOS ONE

Journal Requirements:

Reviewers' comments:

Reviewer's Responses to Questions

**Comments to the Author**

1. Is the manuscript technically sound, and do the data support the conclusions?

Reviewer #1: Yes

Reviewer #2: Yes

2. Has the statistical analysis been performed appropriately and rigorously? 

Reviewer #1: Yes

Reviewer #2: Yes

3. Have the authors made all data underlying the findings in their manuscript fully available?

Reviewer #1: Yes

Reviewer #2: Yes

4. Is the manuscript presented in an intelligible fashion and written in standard English?

Reviewer #1: Yes

Reviewer #2: No

5. Review Comments to the Author

Reviewer #1: I enjoyed reading your paper and this topic sounds interesting and can be potential of publication. Both major and minor issues for improvement are provided. Before resubmitting, please consider the following:

1. How did the findings advance the existing/ previous theory(ies)? The authors need to add contributions to theory and literature explaining how this research fills or advances the existing theories.

2. In the introduction, the theoretical contribution should be clearly indicated. Please explain to which area of knowledge you are contributing. It is not clear yet! The manuscript, in my opinion, also has some practical implications that should also be pointed out there.

It is worth referring to the following studies:

Jum’a, L., Zimon, D., & Ikram, M. (2021). A relationship between supply chain practices, environmental sustainability and financial performance: evidence from manufacturing companies in Jordan. Sustainability, 13(4), 2152.

Gupta, S., & Dutta, K. (2011). Modeling of financial supply chain. European journal of operational research, 211(1), 47-56.

etc.

3. Unfortunately, the limitations of the study were not included in the Conclusions section, though they should be because they certainly exist.

4. It does not demonstrate its innovation and contribution with the comparison of the existing studies, which is a most important key for possible publication. 

Good Luck!

Reviewer #2: This paper selects the panel data of A-share listed companies in China from 2009 to 2021, uses customer concentration and supplier concentration to measure supply chain concentration, and examines the impact of supply chain concentration on enterprise financialization. The results show that both customer concentration and supplier concentration can significantly promote the financialization of enterprises, while inhibiting the substantiation of enterprises. The conclusion is still valid after robustness and endogenous treatment; The analysis of influence mechanism shows that customer concentration affects enterprise financialization by increasing business credit, while supplier concentration has no similar intermediary effect; The results of heterogeneity analysis show that the impact of supply chain concentration on enterprise financialization is significantly different in state-owned enterprises and small and medium-sized enterprises, and more significant in enterprises with less fierce industry competition and located in the central and western regions. The research in this paper provides empirical evidence for restraining the adverse impact of enterprise financialization on industry and taking advantage of the benefit effect brought by supply chain concentration. The aim of the analysis should be evidenced in the abstract and introduction sections. The methodology should be further explained. The results should be discussed in terms of policy implications. The conclusions should be improved with the weaknesses of the analysis and the insights for future research. Finally, the manuscript should be English proofread because some sentences are not clear.

6. PLOS authors have the option to publish the peer review history of their article (what does this mean?). If published, this will include your full peer review and any attached files.

Reviewer #1: No

Reviewer #2: No

---

## [Decision Letter · Decision Letter 1]

20 Apr 2023

Supply Chain Concentration and Enterprise Financialization: Evidence from Listed Companies in China's Manufacturing Industry

PONE-D-22-30905R1

Dear Dr. Zuo,

We’re pleased to inform you that your manuscript has been judged scientifically suitable for publication and will be formally accepted for publication once it meets all outstanding technical requirements.

Kind regards,

Chaohai Shen

Academic Editor

PLOS ONE

Additional Editor Comments (optional):

Reviewers' comments:

Reviewer's Responses to Questions

**Comments to the Author**

1. If the authors have adequately addressed your comments raised in a previous round of review and you feel that this manuscript is now acceptable for publication, you may indicate that here to bypass the “Comments to the Author” section, enter your conflict of interest statement in the “Confidential to Editor” section, and submit your "Accept" recommendation.

Reviewer #1: All comments have been addressed

Reviewer #2: All comments have been addressed

2. Is the manuscript technically sound, and do the data support the conclusions?

Reviewer #1: Yes

Reviewer #2: Yes

3. Has the statistical analysis been performed appropriately and rigorously? 

Reviewer #1: Yes

Reviewer #2: Yes

4. Have the authors made all data underlying the findings in their manuscript fully available?

Reviewer #1: Yes

Reviewer #2: Yes

5. Is the manuscript presented in an intelligible fashion and written in standard English?

Reviewer #1: Yes

Reviewer #2: Yes

6. Review Comments to the Author

Reviewer #1: Dear Authors,

The required changes have been included in the paper. I have no more comments.

Good Luck!

Reviewer #2: The paper has been improved according to the reviewers' comments. Now the manuscript can be accepted for publication.

7. PLOS authors have the option to publish the peer review history of their article (what does this mean?). If published, this will include your full peer review and any attached files.

Reviewer #1: No

Reviewer #2: No

---

## [Editor Report · Acceptance letter]

28 Apr 2023

PONE-D-22-30905R1 

Supply Chain Concentration and Enterprise Financialization: Evidence from Listed Companies in China's Manufacturing Industry 

Dear Dr. Zuo:

I'm pleased to inform you that your manuscript has been deemed suitable for publication in PLOS ONE. Congratulations! Your manuscript is now with our production department. 

Kind regards, 

on behalf of

Dr. Chaohai Shen 

Academic Editor

PLOS ONE